

# Development and characterization of fourteen novel microsatellite markers for the chestnut short-tailed fruit bat (*Carollia castanea*), and cross-amplification to related species

Katherine A. Cleary[1], Lisette P. Waits[1] and Paul A. Hohenlohe[2]

[1] Department of Fish and Wildlife Sciences, University of Idaho, Moscow, ID, United States
[2] Department of Biological Sciences, Institute of Bioinformatics and Evolutionary Studies, University of Idaho, Moscow, ID, United States

## ABSTRACT

Rapid anthropogenic land use change threatens the primary habitat of the Chestnut short-tailed bat (*Carollia castanea*) throughout much of its range. Information on population genetic structure can inform management strategies for this widespread frugivorous bat, and effective protection of *C. castanea* will also benefit the more than 20 mutualistic plant species of which this bat is the primary seed disperser. To facilitate understanding of population genetic structure in this species, fourteen novel microsatellite markers were developed using restriction-site-associated DNA libraries and Illumina sequencing and tested on 28 individuals from 13 locations in Costa Rica. These are the first microsatellite markers developed for *C. castanea*. All loci were polymorphic, with number of alleles ranging from 2–11 and average observed heterozygosity of 0.631. Markers were also cross-amplified in three additional frugivorous bat species threatened by habitat loss and fragmentation: Sowell's short-tailed bat (*Carollia sowelli*), Seba's short-tailed bat (*Carollia perspicillata*), and the Jamaican fruit bat (*Artibeus jamaicensis*), and 10, 11, and 8 were polymorphic, respectively.

Corresponding author
Katherine A. Cleary,
katecleary98@gmail.com

## INTRODUCTION

The Chestnut short-tailed bat (*Carollia castanea*) is a frugivorous bat which inhabits tropical forests from Honduras to Bolivia, and is the primary seed disperser of many pioneer plant species found in regenerating forests (*Lopez & Vaughan, 2007*). Throughout this species' range, rapid conversion of native forest cover to agriculture is driving habitat loss and fragmentation, which threatens the ability of *C. castanea* populations to maintain genetic connectivity. *C. castanea* has a small body size, limited home range of >7 ha, and low wing loading, all of which are associated with lower vagility and increased vulnerability to fragmentation in bats (*Bonaccorso et al., 2006*; *Meyer et al., 2008*; *Meyer, Kalko & Kerth, 2009*). As a result, populations of *C. castanea* in fragmented agricultural landscapes are at

risk of interrupted gene flow, genetic drift, and inbreeding, which reduce genetic diversity and adaptive capacity in the face of future perturbations (*Willi et al., 2007*; *Méndez, Tella & Godoy, 2011*).

To date, only one study has used molecular markers to evaluate the impact of ongoing land use change on *C. castanea*. *Ripperger et al. (2014)* sequenced the mitochondrial d-loop of 173 *C. castanea* individuals sampled from 10 plots in continuous forest and remnant forest patches in a fragmented agricultural landscape in northern Costa Rica, and found no evidence of significant genetic structure. However, mitochondrial markers have a much slower mutation rate than nuclear DNA markers, and are thus less capable of detecting recent response to landscape change (*Wang, 2010*). Since forest fragmentation and agricultural expansion in tropical regions have happened very recently on an evolutionary timescale, we would expect that only neutral nuclear DNA markers with high rates of mutation such as microsatellites would already show a response to these processes. Despite the utility of microsatellite markers for evaluating population responses to recent land use change, no microsatellite loci have previously been developed for *C. castanea*. In fact, of the seven currently recognized species in the *Carollia* genus (*Velazco, 2013*), microsatellite markers have only been developed for one, *Carollia brevicauda* (*Bardeleben et al., 2007*); these authors also successfully cross-amplified eight microsatellites to *C. castanea*.

The goal of this study was to develop the first microsatellite markers specifically for *C. castanea* using Illumina high-throughput sequencing, to fully characterize these markers using a small sample of individuals from Costa Rica, and to evaluate the transferability of these microsatellites to three other co-distributed bat species: Sowell's short-tailed bat (*Carollia sowelli*), Seba's short-tailed bat (*Carollia perspicillata*), andthe Jamaican fruit bat (*Artibeus jamaicensis*). These novel markers can be used to quantify population genetic structure, identify populations that have become genetically isolated due to habitat loss and fragmentation, and evaluate correlations between genetic diversity, gene flow, and land use in fragmented agricultural landscapes. Future studies can also use these markers to increase understanding of mating and dispersal strategies in *C. castanea*; preliminary evidence of female-biased dispersal has been found in this species using mitochondrial DNA markers (*Ripperger et al., 2014*), which is very rare in mammals (*Greenwood, 1980*).

## MATERIALS AND METHODS

Samples of *C. castanea* were collected in 13 remnant forest patches in the San Juan-La Selva biological corridor in northern Costa Rica (Table 1), where conversion of tropical lowland forest to agriculture has led to widespread habitat loss and fragmentation. Bats were captured using mist nets, and tissue samples of the uropatagium were collected using a 2 mm diameter circular biopsy tool and stored in a 2mL tube with lysis buffer (50 mM Tris pH 8.0, 50 mM EDTA, 50 mM sucrose, 100 mM NaCl, 1% SDS) (*Faure, Daniel & Clare, 2009*). Our capture and handling procedures were approved by the University of Idaho's Animal Care and Use Committee (protocol #2011-31), and all field work was conducted with permission from the Costa Rican Ministry of Energy and the Environment (permit number: R-005-2013-OT-CONAGEBIO).

**Table 1** Number of samples of each species collected from the remnant forest patches in the San Juan-La Selva biological corridor in northern Costa Rica. Latitude and longitude of each patch are given in decimal degrees.

| Patch | No. samples C. castanea | No. samples C. sowelli | No. samples C. perspicillata | No. samples A. jamaicensis | Lat | Long |
|---|---|---|---|---|---|---|
| 1 | 1 | 0 | 1 | 0 | 10.6624 | −84.1625 |
| 2 | 1 | 0 | 0 | 0 | 10.4428 | −84.1080 |
| 3 | 1 | 0 | 1 | 0 | 10.4342 | −84.1285 |
| 4 | 2 | 0 | 0 | 1 | 10.4076 | −84.1516 |
| 5 | 1 | 0 | 0 | 0 | 10.4617 | −84.1537 |
| 6 | 2 | 0 | 0 | 0 | 10.4543 | −84.3240 |
| 7 | 1 | 0 | 0 | 0 | 10.4110 | −84.2458 |
| 8 | 2 | 1 | 1 | 0 | 10.4304 | −84.0931 |
| 9 | 1 | 0 | 0 | 1 | 10.5466 | −84.1698 |
| 10 | 2 | 0 | 0 | 0 | 10.5874 | −84.1600 |
| 11 | 1 | 0 | 0 | 0 | 10.5565 | −84.1816 |
| 12 | 3 | 0 | 0 | 0 | 10.5348 | −84.1482 |
| 13 | 2 | 1 | 0 | 0 | 10.4313 | −84.0712 |

Genomic DNA was extracted from tissue samples of three individuals using the Qiagen Blood and Tissue Kit. Libraries were prepared using a restriction-site-associated DNA approach (*Etter et al., 2011*). In brief, genomic DNA was digested with a restriction enzyme, and an adapter containing a 6 bp long RAD tag and both forward amplification and Illumina sequencing priming sites was ligated to the fragments. All fragments were pooled, sheared, and size selected. A second adapter was then ligated to the size selected fragments; this adapter is designed to ensure that only P1 adapter-ligated RAD tags will be amplified during the final amplification step. Final prepared libraries were run on one lane of an Illumina® MiSeq250, which generated 3,179,284 250-bp sequences. Sequences were archived in the NCBI Sequence Read Archive (accession # SRP082144). Data was de-multiplexed and quality-cleaned using Stacks V.1.21 (*Catchen et al., 2013*). A total of 2,300,295 cleaned sequences were run through the program QDD V.3.1 (*Meglécz et al., 2014*) to identify microsatellites, filter out redundant sequences, and design primers. This process identified 10, 558 sequences containing at least one microsatellite.

All sequences were screened using strict criteria to select only perfect microsatellites with di- or tetranucleotide motifs, at least five repeats, and low alignment scores with known transposable elements. From the 656 candidate loci identified with these criteria we selected 32 high-quality loci to test for amplification. Unlabeled forward and reverse primers for these loci were synthesized through Applied Biosystems. Primers were diluted to a 10uM solution containing both forward and reverse primers, and tested on eight *C. castanea* individuals from eight different remnant forest patches. Individual amplifications were performed in a 7 μL reaction containing 2 μL template DNA (at 7 ng/μL), 2X Qiagen Multiplex PCR Master Mix, 0.5X Q solution, and 0.10 μL of each 10 μM primer solution. Cycling conditions consisted of a 15 min initial denaturation at 95 °C, followed by 15

cycles of a touchdown protocol of 94 °C for 30s; 63 °C for 90 s; 72 °C for 60 s, and then 20 additional cycles of 94 °C for 30 s; 57 °C for 90 s; 72 °C for 60 s.

Amplification products were examined for polymorphism using standard gel electrophoresis with 3% agarose gels. Twenty loci were identified as potentially polymorphic. For these twenty loci, fluorescent labeled forward primers and unlabeled reverse primers were synthesized through Integrated DNA technologies and Applied Biosystems. Using the same amplification reactions and cycling conditions as for the previous step, these loci were amplified and amplification products were separated on an Applied Biosystems 3130xl Analyzer with LIZ500 internal size standard, and scored using GeneMapper 5 (Applied Biosystems). Fourteen of the twenty loci were identified as definitively polymorphic. These loci were multiplexed into two reactions (Table 2) and tested on 20 new *C. castanea* individuals. To obtain estimates of population genetic parameters representative of the study area, we selected these individuals from thirteen different patches across the study area: the same eight remnant forest patches from the previous step, plus an additional five patches (Table 1). Cycling conditions were the same as used above for both multiplexes. Amplification products were separated on an Applied Biosystems 3130xl Analyzer with LIZ500 internal size standard, and scored using GeneMapper 5 (Applied Biosystems). To validate scoring methods, the distribution of raw allele sizes was visualized and the best bin sets for each locus were generated using Autobin v.09 (Fig. S1). All loci were tested twice using DNA from the same *C. castanea* samples to ensure reliable results.

Number of alleles per locus ($N_A$), observed heterozygosity ($H_O$), expected heterozygosity ($H_E$), and tests for departures from Hardy-Weinberg equilibrium ($P_{\mathrm{HWE}}$) were calculated in GenAlEx 6.5 (*Peakall & Smouse, 2012*), and loci were screened for null alleles in CERVUS 3.0.7 (*Kalinowski, Taper & Marshall, 2007*). To assess baseline frequencies of each allele across the study area, GenAlEx 6.5 was used to calculate allele frequencies across all samples. Next, program STRUCTURE v 2.3.4 (*Pritchard, Stephens & Donnelly, 2000*) was used to test for genetic structure in the data. We chose an admixture model with correlated allele frequencies; this model is appropriate for our system because since land use change in the study area is a recent event we expected that allele frequencies in the remnant forest patches would still be fairly similar (*Falush, Stephens & Pritchard, 2003*). Since the samples were collected from 13 different forest patches and it is possible that *C. castanea* populations in these patches represent 13 distinct genetic groups, we tested all values of K between one and 13. As recommended by *Gilbert et al. (2012)*, we ran the model for 100,000 generations, with a 100,000 generation burn-in, and confirmed this number of generations was adequate by checking for convergence of alpha, F, D, and log likelihood. We ran three independent replicate runs using the same model settings.

All fourteen loci were also tested for amplification and polymorphism in two individuals of *Carollia sowelli*, three individuals of *Carollia perspicillata*, and three individuals of *Artibeus jamaicensis*. These species were chosen because like *C. castanea* they face threats from habitat destruction throughout their range and together are key seed dispersers for hundreds of species of Neotropical plants (*Ortega & Castro-Arellano, 2001*; *Thies & Kalko, 2004*, *Lopez & Vaughan, 2007*). Samples from all three species were collected in the same remnant forest patches where the *C. castanea* samples were collected (Table 1), and under the same handling and collection permits as described above. In the laboratory, the PCR

**Table 2 Microsatellite loci characteristics.** Microsatellite loci developed and characterized in 28 Chestnut short-tailed fruit bat (*C. castanea*) samples from Costa Rica. Fluorescent labels attached to forward primers are in brackets.

| Locus | GenBank accession no. | Repeat motif | Primer 5′–3′ | Range (bp) | MP | NA | $H_O$ | $H_E$ | $P_{HWE}$ |
|---|---|---|---|---|---|---|---|---|---|
| CC-7 | KX060618 | (AC)13 | [PET]GAGTAACAAATAAGAGGGAACTGGG GCAACTGCTCACAACCTGTT | 292–300 | 1 | 5 | 0.800 | 0.715 | 0.385 |
| CC-10 | KX060619 | (AATG)7 | [FAM]TGCAGGGAAGATGAGAATGAACA CAGGGCCTGGTGCATAGTAG | 116–128 | 1 | 4 | 0.450 | 0.431 | 0.981 |
| CC-12 | KX060620 | (ACATAT)12 | [VIC]ACAGACCAAGAACAGAGCTG ATGATCTCTGAGCGCTCACA | 236–420 | 1 | 11 | 0.929 | 0.870 | 0.389 |
| CC-13 | KX060621 | (AG)6 | [NED]CCGAGTCGTTTAGGCTGGTT GCCCAACCCTGTCTTTGTC | 181–185 | 1 | 2 | 0.500 | 0.455 | 0.658 |
| CC-18 | KX060622 | (AAGG)13 | [PET]AGCAGGACGTAAGACAGCAG TTCCATTTCATTGCTGTGGC | 234–245 | 1 | 4 | 0.632 | 0.622 | 0.159 |
| CC-19 | KX060623 | (AC)18 | [PET]CCCTGCACCAAATCAGCAAT CTGCCAGCAATGCGTGAATG | 120–142 | 1 | 6 | 0.650 | 0.703 | 0.558 |
| CC-20 | KX060624 | (AT)11 | [VIC]AGGAAGGGAGTCACCATGGT CCAACCAGGTGTTAGTGCTA | 178–226 | 2 | 8 | 0.550 | 0.700 | 0.257 |
| CC-23 | KX060625 | (AG)21 | [NED]CCTTCTATCTGTGACGCTGCT TCACGCAACAAACAGTAAGTGA | 226–256 | 1 | 10 | 0.750 | 0.781 | 0.898 |
| CC-24 | KX060626 | (ACAG)5 | [NED]GCAGGACAGGGAGCTTGAAA ATCATAGAAAGTCGCTGTTGCT | 136–140 | 2 | 2 | 0.368 | 0.494 | 0.267 |
| CC-25 | KX060627 | (AATG)8 | [NED]GTCTGTTTCTGCCTCTTTGGG ATGGGTCACCGTGTCTTAGC | 129–141 | 1 | 4 | 0.600 | 0.554 | 0.932 |
| CC-26 | KX060628 | (AC)22 | [FAM]GAGGTACGCAGCCAGATGTG ACTGCTTTCTGGTGCTTCTCA | 236–256 | 1 | 11 | 0.900 | 0.866 | 0.813 |
| CC-27 | KX060629 | (AC)21 | [FAM]GCAGGGAGTGGAGCATCATC TGTTGCCAGGTTGTCACAGT | 193–209 | 1 | 9 | 0.750 | 0.776 | 0.892 |
| CC-29 | KX060630 | (AC)12 | [VIC]ACCCTTGCTAGTCTGCCAAC GAAGGCTCGGTCCTGCTC | 220–230 | 1 | 6 | 0.850 | 0.785 | 0.660 |
| CC-30 | KX060631 | (AGGG)7 | [VIC]AGGCAAACCCACAGACCAAA CCAGTCTGTTCTCATTCCCGT | 119–131 | 2 | 3 | 0.100 | 0.329 | 0.003 |

**Notes.**
MP, Multiplex locus was assigned to; NA, Number of alleles per locus; $H_e$, Expected heterozygosity; $H_o$, Observed heterozygosity; $P_{HWE}$, Probability the locus is in 2 Hardy-Weinberg equilibrium.

conditions were the same as used for *C. castanea*, but loci were tested separately to avoid the potential problem of overlapping alleles caused by shifting size ranges in the new species. Population genetic analyses were not conducted on these data due to the small sample size for each species.

## RESULTS AND DISCUSSION

We successfully developed and characterized 14 novel microsatellite markers for *C. castanea*, of which 13 are likely to be useful for future research on this species. In addition, our tests of cross-amplification to *C. sowelli*, *C. perspicillata,* and *Artibeus jamaicensis* revealed that a large subset of these loci amplify and are polymorphic in these species as well.

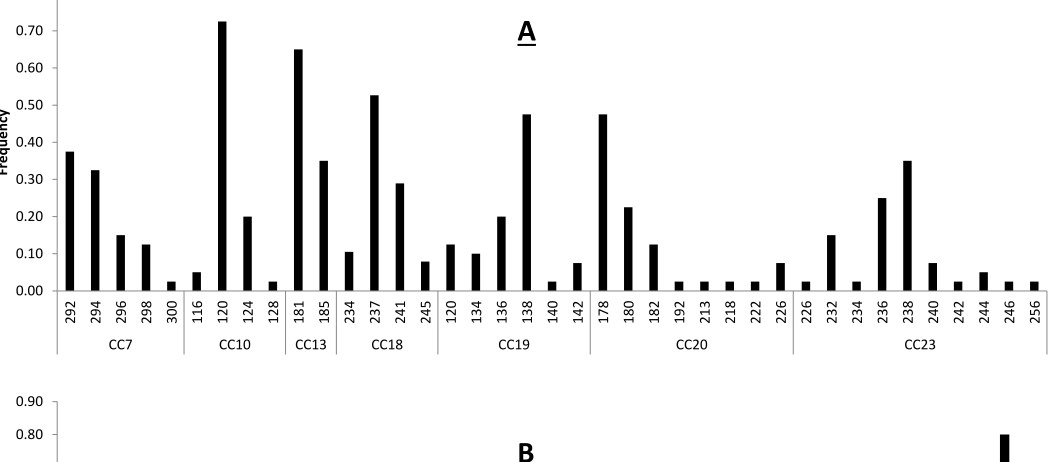

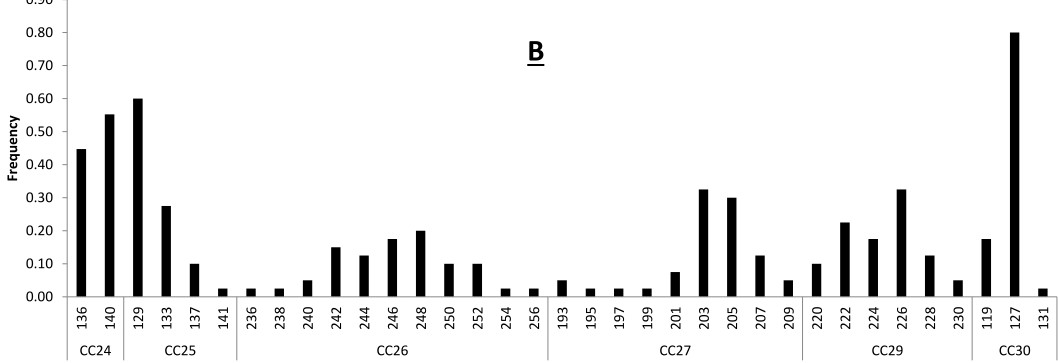

**Figure 1** **Baseline allele frequencies for all loci, averaged across all 20 sampled individuals of *C. cas-tanea*.** Loci 7–23 are shown in (A) and loci 24–30 in (B).

Primer sequences, size range of amplification product, and multiplex assignment for each of the fourteen microsatellite loci are presented in Table 2. All loci were in HWE with the exception of CC-30 ($p = 0.003$) (Table 2), and all loci had null allele frequencies of <1% except CC-24 (14%) and CC-30 (53%). The null allele rate in CC-24 is moderate and this locus was not out of HWE, so we consider it a reliable marker for use in *C. castanea*. The marker CC-30 was significantly out of HWE and showed a fairly high null allele rate in these analyses, so it is not likely that this marker will prove to be reliable for use in this species. Excluding these two potentially problematic loci, remaining loci had 2–11 alleles per locus, with an average observed heterozygosity of 0.631 (±0.227) (Table 2). These levels of polymorphism and heterozygosity are similar to those found by *Bardeleben et al. (2007)*: the loci these authors cross-amplified from *C. brevicauda* to *C. castanea* showed 2–18 alleles per locus, with an average observed heterozygosity of 0.69. Calculation of allele frequencies across the study area revealed that loci CC-10 and CC-30 are the only two loci where a single allele has a frequency of greater than 0.7 (Fig. 1).

For all three independent replicate runs in STRUCTURE, $K = 1$ had the largest log likelihood (closest to zero) of all tested values. This indicated that the most likely number of genetic groups was one ($K = 1$), with a posterior probability the first run of ln Pr $(X/K) = 0.49$. These results are in line with the findings from *Ripperger et al. (2014)*, who used mitochondrial DNA markers and found no significant genetic structure among

**Table 3  Cross amplification in related species.** Cross-species amplification success, range in base pairs, and number of alleles (NA) for novel *C. castanea* loci in three related Phyllostomid species.

| Locus | *Carollia sowelli* | | | *Carollia perspicillata* | | | *Artibeus jamaicensis* | | |
|---|---|---|---|---|---|---|---|---|---|
| | Success | Range (bp) | NA | Success | Range (bp) | NA | Success | Range (bp) | NA |
| CC-7 | 2/2 | 286–296 | 3 | 3/3 | 286–296 | 4 | 3/3 | 353–381 | 5 |
| CC-10 | 2/2 | 124–146 | 3 | 3/3 | 112–146 | 6 | 3/3 | 112–138 | 4 |
| CC-12 | 0/2 | – | – | 0/3 | – | – | 0/3 | – | – |
| CC-13 | 2/2 | 153–163 | 2 | 3/3 | 153–163 | 2 | 0/3 | – | – |
| CC-18 | 2/2 | 234–264 | 4 | 3/3 | 210-292 | 5 | 1/3 | 276–276 | 1 |
| CC-19 | 2/2 | 124–132 | 2 | 2/3 | 122–132 | 3 | 2/3 | 116–122 | 3 |
| CC-20 | 2/2 | 193–263 | 4 | 3/3 | 165–221 | 6 | 3/3 | 160–162 | 2 |
| CC-23 | 0/2 | – | – | 0/3 | – | – | 0/3 | – | – |
| CC-24 | 0/2 | – | – | 2/3 | 125–133 | 2 | 3/3 | 124-124 | 1 |
| CC-25 | 2/2 | 135–145 | 3 | 3/3 | 135–145 | 4 | 0/3 | – | – |
| CC-26 | 2/2 | 215–217 | 2 | 3/3 | 215–219 | 3 | 0/3 | – | – |
| CC-27 | 2/2 | 185–197 | 2 | 3/3 | 185–199 | 3 | 3/3 | 188–192 | 3 |
| CC-29 | 2/2 | 229–233 | 2 | 3/3 | 213–239 | 4 | 3/3 | 189–203 | 2 |
| CC-30 | 0/2 | – | – | 0/3 | – | – | 0/3 | – | – |

*C. castanea* populations in the same region. However, it is important to note that since the primary purpose of the present study is to develop new microsatellite markers, only a very small number of individuals were sampled ($n = 20$). It is possible that analyzing additional individuals and individuals from more isolated forest patches could reveal significant genetic structure at the scale of the study area.

The novel microsatellite markers we have developed here will facilitate such future studies of population genetic structure in *C. castanea* and enable tests of whether levels of genetic diversity and gene flow in isolated populations are correlated with land use change and habitat loss and fragmentation. Understanding the impact of these processes on *C. castanea* is especially important since this bat is known to disperse at least 20 species of Neotropical plants (*Lopez & Vaughan, 2007*). If *C. castanea* is able to maintain gene flow in fragmented landscapes, then these mutualistic plant species will also have a better chance of maintaining reproductive connectivity, genetic diversity, and recolonization capacity. In addition, these markers can be used to help resolve persistent taxonomic uncertainty within the *Carollia* genus (*Velazco, 2013*). Previous studies have used mitochondrial DNA markers to identify cryptic species within *C. castanea*, including *C. benkeithi* in Ecuador and Panama (*Solari & Baker, 2006*), and an unnamed species from samples collected in Panama (*Velazco, 2013*). Although we are confident that all of the samples used in this study are *C. castanea* since neither of these cryptic species overlaps in range with our study area, this taxonomic uncertainty should be considered when using the microsatellite markers presented here.

Our analyses were also successful in determining the utility of these microsatellite markers in the three co-distributed frugivorous bat species. Ten loci amplified and were polymorphic for *C. sowelli*; these represent the first microsatellite markers available for

this species (Table 3). Eleven loci amplified and were polymorphic for *C. perspicillata*, and 8 loci amplified for *A. jamaicensis*, but only 6 were polymorphic (Table 3). Rates of polymorphism for these loci in these three species may be higher than reported here, since loci were tested in a small number of individuals of each species ($n = 2$–3 individuals per species). In addition, allele frequencies reported in Table 3 may not be representative of allele frequencies at these loci in the larger populations, since testing loci in only a few individuals can lead to ascertainment bias. Although microsatellite markers have previously been cross-amplified to *C. perspicillata* and directly developed for *A. jamaicensis*, these additional markers will add resolution and power to future studies of genetic patterns in these species.

## ACKNOWLEDGEMENTS

The authors thank the landowners of Sarapiquí for allowing access to their land and Henry Lara Perez for help with species identification and sample collection. We are grateful to Cody Weinch and Tamara Max for invaluable assistance with developing RAD libraries.

### Funding

This research was supported by the NIH National Institute of General Medical Sciences under grant P30 GM103324, by the National Science Foundation under IGERT grant 0903479 and CNH grant 1313824, and by Bat Conservation International. KAC also received support from the Fulbright Student Scholarship Program to collect field data in Costa Rica. The funders had no role in study design, data collection and analysis, decision to publish, or preparation of the manuscript.

### Grant Disclosures

The following grant information was disclosed by the authors:
NIH National Institute of General Medical Sciences: P30 GM103324.
National Science Foundation: 0903479, 1313824.
Fulbright Student Scholarship Program.
Bat Conservation International.

### Competing Interests

The authors declare there are no competing interests.

### Author Contributions

- Katherine A. Cleary conceived and designed the experiments, performed the experiments, analyzed the data, wrote the paper, prepared figures and/or tables.
- Lisette P. Waits and Paul A. Hohenlohe conceived and designed the experiments, contributed reagents/materials/analysis tools, reviewed drafts of the paper.
## Animal Ethics

The following information was supplied relating to ethical approvals (i.e., approving body and any reference numbers):

Animal Care and Use Committee Protocol #2011-31—Ecology and Conservation Genetics of Phyllostomid Bats in the Human-Dominated Landscape of San Juan-La Selva Biological Corridor.

## Field Study Permissions

The following information was supplied relating to field study approvals (i.e., approving body and any reference numbers):

Costa Rican Ministry of Energy and the Environment, Permit number: R-005-2013-OT-CONAGEBIO.

## DNA Deposition

The following information was supplied regarding the deposition of DNA sequences:

Sequence data is archived at NCBI Sequence Read Archive, accession #: SRP082144.

Microsatellite primer sequences are archived in NCBI GenBank, accession numbers: KX060618, KX060619, KX060620, KX060621, KX060622, KX060623, KX060624, KX060625, KX060626, KX060627, KX060628, KX060629, KX060630, KX060631.

## Data Availability

NCBI SRA: SRP082144.

## Supplemental Information

Supplemental information for this article can be found online at http://dx.doi.org/10.7717/peerj.2465#supplemental-information.

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
