# Peer review of "Development and characterization of fourteen novel microsatellite markers for the chestnut short-tailed fruit bat (Carollia castanea), and cross-amplification to related species"

_PeerJ, doi:10.7717/peerj.2465_

## Round 0.1 · original submission · Minor Revisions

This manuscript on the isolation and characterization of 14 microsatellite markers in a fruit bat has now been reviewed by three Referees with expertise in bat ecology and population genetics. As the consensus among Referees suggests, such a primer note would be of interest to readers of Peer J following revision. In particular, the sequence data needs to be easily accessible in a format that is generally accepted for such data (e.g., NCBI). There should also be an ethical statement included about animal care. Finally, clarifications on the methodology along with the helpful suggested edits to be clear on scope are warranted to improve the paper. Thank you for submitting your work to PeerJ.

·

Basic reporting

No Comments

Experimental design

No Comments

Validity of the findings

No Comments

Additional comments

The manuscript by Cleary et al. is a short methodological paper reporting the development and characterization of 14 microsatellites markers in a fruit bat [of which 12 to13 meet the standards for future use in a population genetic analysis-CC-30 (and perhaps CC-24) should be discarded]. Overall, the paper is well written and the quality of the markers was correctly assessed (i.e., null alleles, polymorphism, HWE). My only major suggestion is that I usually find it useful to visualize the distribution of raw allele size for each microsatellite markers (for an example see Lefèvre et al. 2012). These can easily be generated using Autobin (http://www6.bordeaux-aquitaine.inra.fr/biogeco/Production-scientifique/Logiciels/Autobin). I find it useful to 1) visualize how the allele sizes are distributed -was the binning into discrete alleles really obvious or were the allele sizes rather continuous?-; 2) provide a baseline for the frequency of each allele. Sampling different populations might likely end up with different patterns of allelic frequencies but I feel it is good practice to present the distribution when documenting SSR markers for the first time. This is not mandatory but it will certainly contribute to beef up the manuscript.

Minor comments
Line 81: Replace Table 4.1 with Table 1
Line 92: One could argue that 10,000 MCMC chain is unusually too low. Yet, I admit this is not a full-fledged population genetics analysis you aimed to conduct here. Hence I assume these parameters might be fine for such exploratory purpose.
Line 98: Replace Table 4.1 with Table 1
Line 100: Replace Table 4.1 with Table 1
Line 100: Table 1 does not report CERVUS (null allele frequencies) results. Add the results to the table or do not refer to the table when reporting these values in the main text.
Lines 101-105: You might want to clearly advise readers to remove CC-30 (and perhaps CC-24 if you want to be more conservative with regards to the probability of null alleles). This is probably the best solution: because these two loci are amongst the least polymorphic (low number of alleles: 3 and 2, respectively) few information will be lost by removing them, yet the remaining available information will be more reliable (less probability of null alleles).
Line 105: Replace Table 4.1 with Table 1
Lines 116-122: Move to the Material and Methods section and please provide the sampling size for each species.
Lines 123-133: The numbers reported in this paragraph do not match does reported in Table 2. In table 2, I count 10 polymorphic loci (not 11) for C. sowelli; 11 polymorphic loci (not 12) for C. perspicillata and I agree that there 8 loci amplified (of which 6 were polymorphic) for A. jamaicensis. Please revise or explain.
Line 129: Replace Table 4.2 with Table 2
Line 131: add a parenthesis specifying the low sample size for these species i.e. "small number of individuals assayed of each species (n = 2-3 sampled individuals per species)."
Abstract: In the last sentence of the abstract, you report 11, 12, and 8 polymorphic loci (as reported in the main text at lines 123-133). Relying on Table 2, I expect the right numbers should be 10, 11, and 6 polymorphic loci (or 10,11, and 8 loci that were successfully amplified).

·

Basic reporting

I have two concerns about PeerJ policies. The first is related to the submission of raw sequence data to public databases. The authors provide a link to Dropbox, which is not the most suitable choice for sequence data. I would suggest the authors use the SRA database on NCBI, for instance. Second, one of the policies states that "Manuscripts that report the characterization of specific primers (such as microsatellites) should include substantial biological analyses." While the manuscript is successful in reporting new SSR loci for C. castanea, I found the biological analysis (in this case, for instance, genetic diversity and structure of C. castanea populations) preliminary. Results on genetic structure are not discussed or compared to available literature.

I believe the Introduciton section would benefit from a review of the use of molecular markers in population genetics of Carollia as a genus or of Carollia castanea in particular. If these are the first SSR markers developed for the species, were other types of markers ever used?

Experimental design

The research question is not clearly stated, even though the knowledge gap is identified.

The Methods section does not include the Institution Review Board mentioned in the Declaration about work involving vertebrate animals. It does not include the permit number for field study either.

The methods section lacks information for reproducibility of results (please see point-by-point comments below).

Validity of the findings

Once again, no accession codes for SSR loci or short-read data are mentioned in the text. The Dropbox link sent in the submission form is not an acceptable repository for sequence data.

No conclusions were stated in the manuscript.

Additional comments

Below are some general comments, and questions raised while reading the manuscript.

Introduction

I believe the Introduciton section would benefit from a review of the use of molecular markers in population genetics of Carollia as a genus or of Carollia castanea in particular. If these are the first SSR markers developed for the species, were other types of markers ever used?

Materials and methods

The Methods section does not include the Institution Review Board mentioned in the Declaration about work involving vertebrate animals. It does not include the permit number for field study either.

Line 49: How many samples were collected from each location? Do authors have their GPS locations? While the sample size used in this study seems reasonable for marker development and validation, is it adequate for analyses of population structure and diversity in bats? Do authors plan on increasing sample size in future studies?

Line 53: Please describe contents of lysis buffer with concentrations.

Lines 55-56: I could not read the protocol described in Etter et al (2011) because it was behind a paywall. Please describe how libraries were prepared for RAD-sequencing.

Line 56: Do the authors mean they used an Illumina MiSeq with 250-bp reads?

Line 56-58: Were these reads submitted to any public database such as the Short Read Archive at NCBI? Will they be kept only in Dropbox?

Lines 59-60: Does this process include any assembly step, or only filtering out of redundant sequences?

Lines 61-63: How "strict" were the criteria for a low alignment score with transposable elements? How were the sequences aligned to known transposable elements? Which database, if any, was used? Were they known in bats? Mammals?

Line 63: Were these loci submitted to any public databases? I could see they were from your Declarations in the manuscript submission form. Please include the NCBI accession numbers in the manuscript.

Lines 68-69: Please mention the DNA concentration of samples used for PCR, and not their volume.

Lines 81-82: Do these 13 locations include those eight that were previously used to screen polymorphic loci? Are they also referenced with GPS data? How many samples from each location were used in this step?

Line 86: Did authors perform two DNA extractions from the same C. castanea samples or were these two reactions performed with DNA from a single extraction?

Line 92: The burn-in size (10,000) and the number of iterations (10,000) seem extremely small. What led the authors to set these parameters? For each K, did authors perform replicate runs? Most importantly, with these parameters, did summary statistics such as alpha, F, D, and likelihood converge adequately? Please refer to Gilbert et al (Molecular Ecology, 2012, 21:4925-4930) for recommendations of good practices for reporting methods and results of analysis with Structure. These are extremely important for reproducibility (more comments below).

Results and Discussion

Line 105-106: What criteria was used to select the most likely number of genetic groups from Structure?

Lines 123-126: I believe these lines should be in the Materials and Methods section.

Lines 108-115: Is this the first study on population genetics of C. castanea? Were other types of markers ever used for this purpose on this species? How do these results compare to those found by Bardeleben et al. 2007 regarding number of alleles or other statistics such as observed and expected heterozygosity?

Conclusion: What is the conclusion of your work?

·

Basic reporting

There was a field component where wild bats were captured and tissue samples collected however, there is no mention of IACUC or permits that were obtained. As per Peer J policies an ethics statement is required in the Materials and Methods for field studies and collecting of tissue samples.

Experimental design

no comments

Validity of the findings

No comments

Additional comments

Line 63 How many candidate loci were left after applying your strict criteria?
Why did you select only 32 to test for amplification? It seems there was an opportunity here to get a lot more loci, say 50 or so, which could really help with inform the taxonomic issues within this species (see PM Velazco 2013 Zootaxa 3718(3) 267-278). See more about this in comments below.

Line 108 I would change this to either 12 or 13. Likely, no one will chose to use CC-30.

Line 112 add "seeds" after "Netotropical plants"

Lines 129-131, this may also be ascertainment bias, which should be acknowledged.

It is a bit concerning that these markers were not tested on any sister taxa or on samples of C. castanea from other parts of its range as it has been demonstrated that this is a species complex with cryptic lineages. I understand the limitations of sampling in other countries or getting samples from museums but I feel it needs to be addressed that these markers have been designed in a species that has very uncertain taxonomy currently. It could be argued that these markers could help with these issues or that they may be insufficient because we have no idea how they will work for this species complex. I would at least like to see this issue addressed as these markers can not be assumed to be useful across the species range based on the samples here and the current taxonomic status (PM Velazco 2013 Zootaxa 3718(3) 267-278).

I think there is tremendous value is designing primers for a specific species of interest as problems with ascertainment bias and the use of markers designed in other species is well known. I appreciate that the authors carefully explained the value of these particular species-specific markers. I think it would have been more useful to optimize a higher number of markers particularly for this species but I do not dismiss the value of this work as it stands.

---

## Round 0.2 · accepted · Accept

The authors have done a really great job addressing the comments and concerns of the thorough Referee suggestions. I also appreciate the effort to make the data accessible for other researchers. Thank you for sending your work to PeerJ.